# The Lack of Analgesic Efficacy of Nefopam after Video-Assisted Thoracoscopic Surgery for Lung Cancer: A Randomized, Single-Blinded, Controlled Trial

**DOI:** 10.3390/jcm11164849

**Published:** 2022-08-18

**Authors:** Hyean Yeo, Ji Won Choi, Seungwon Lee, Woo Seog Sim, Soo Jung Park, Heejoon Jeong, Mikyung Yang, Hyun Joo Ahn, Jie Ae Kim, Eun Ji Lee

**Affiliations:** 1Department of Anesthesiology and Pain Medicine, CHA Ilsan Medical Center, CHA University, Goyang 10414, Korea; 2Department of Anesthesiology and Pain Medicine, Samsung Medical Center, Sungkyunkwan University School of Medicine, Seoul 06351, Korea; 3Department of Anesthesiology and Pain Medicine, Seongnam Citizens Medical Center, Seongnam 13290, Korea

**Keywords:** nefopam, video-assisted thoracoscopic surgery, acute postoperative pain, opioid consumption, chronic post-surgical pain, lung cancer

## Abstract

Nefopam is a centrally acting non-opioid analgesic, and its efficacy in multimodal analgesia has been reported. This study aimed to assess the analgesic efficacy of intraoperative nefopam on postoperative pain after video-assisted thoracoscopic surgery (VATS) for lung cancer. Participants were randomly assigned to either the nefopam or the control group. The nefopam group received 20 mg of nefopam after induction and 15 min before the end of surgery. The control group received saline. The primary outcome was cumulative opioid consumption during the 6 h postoperatively. Pain intensities, the time to first request for rescue analgesia, adverse events during the 72 h postoperatively, and the incidence of chronic pain 3 months after surgery were evaluated. Ninety-nine patients were included in the analysis. Total opioid consumption during the 6 h postoperatively was comparable between the groups (nefopam group [*n* = 50] vs. control group [*n* = 49], 19.8 [13.5–25.3] mg vs. 20.3 [13.9–27.0] mg; median difference: −1.55, 95% CI: −6.64 to 3.69; *p* = 0.356). Pain intensity during the 72 h postoperatively and the incidence of chronic pain 3 months after surgery did not differ between the groups. Intraoperative nefopam did not decrease acute postoperative opioid consumption or pain intensity, nor did it reduce the incidence of chronic pain after VATS.

## 1. Introduction

Severe postoperative pain after thoracic surgery interferes with deep breathing and coughing and reduces pulmonary function, which increases the incidence of postoperative pulmonary complications [1,2]. Furthermore, severe pain in acute postoperative periods seems to be strongly associated with the development of chronic post thoracotomy pain syndrome (CPTPS) [3,4,5]. The CPTPS, defined as pain which persists or recurs longer than 3 months after thoracotomy [6], has been reported in its incidence up to 80% of patients at 3 months and 61% 1 year after surgery [1]. The patients with CPTPS suffer from neuropathic or sympathetically mediated pain as well as nociceptive pain, and it can interfere with patients’ daily activities and further reduce their quality of life [3,4,6]. Therefore, several efforts have been made to manage or prevent acute and chronic post-thoracotomy pain. Thoracic epidural analgesia was widely used as the gold standard for pain control. However, it has relatively common complications, such as hypotension and urinary retention, and has low cost effectiveness [7,8]. Opioids are also an important component of pain management; however, the indiscriminate use of opioids can result in adverse events, such as decreased consciousness, delayed mobilization and constipation. Currently, multimodal analgesia is recommended to optimize analgesia and minimize opioid-related side effects [1,9,10].

Nefopam is a centrally acting, non-opioid, non-steroidal analgesic drug, and its efficacy in multimodal analgesia has been reported in some previous studies [11,12]. The mechanism of nefopam is not fully understood; however, the inhibition of serotonin, norepinephrine, and dopamine re-uptake is known to play a main role in its analgesic effect. It also reduces the activity of post-synaptic glutamate receptors, such as N-methyl-D-aspartate (NMDA) receptors, by modulating calcium and sodium channels [12,13,14]. Therefore, we expected that nefopam could decrease postoperative opioids consumption and, based on its mechanisms, contribute to reducing the incidence of CPTPS.

Although the analgesic efficacy of nefopam was reported in several surgeries, including abdominal and orthopedic surgeries [15,16,17,18], it has not been assessed in patients undergoing video-assisted thoracoscopic surgery (VATS). Therefore, this prospective study aimed to evaluate the analgesic efficacy of intraoperative nefopam on acute and chronic postoperative pain after VATS in lung cancer patients. The primary outcome was total opioid consumption during the first 6 h postoperatively. The secondary outcomes were pain intensities, the time to first request for rescue analgesia, adverse events during the 72 h postoperatively, and the incidence of chronic pain evaluated 3 months after surgery. We hypothesized that intraoperative nefopam would reduce postoperative opioid consumption after VATS for lung cancer.

## 2. Materials and Methods

### 2.1. Study Design and Ethical Statements

This prospective, single-blinded, randomized controlled trial was approved by the Institutional Review Board (IRB No: SMC 2020-12-167, approval date: 22 February 2021) and registered with the Korean Clinical Research Information Service (registration No: KCT0006246; principal investigator: Ji Won Choi; date of registration: 11 June 2021; http://cris.nih.go.kr). Screening and enrollment for the study were conducted between March 2021 and September 2021 at a tertiary academic hospital in Seoul, South Korea. Written informed consent was obtained from all participants. This study was performed in accordance with the ethical principles of the 1964 Declaration of Helsinki and its later amendments. The trial was conducted following an original protocol and CONSORT guideline [19].

### 2.2. Participants

Patients between 20 and 70 years of age with ASA physical statuses I to III who were scheduled for elective VATS lung lobectomy were included. The exclusion criteria were as follows: patient refusal to participate, allergy to nefopam, renal dysfunction (serum creatinine > 1.5 mg/dL), hepatic dysfunction, history of seizure or epilepsy, recent myocardial infarction, current use of monoamine oxidase inhibitor, urinary tract disease causing urinary retention, and closed angle glaucoma.

### 2.3. Randomization and Blinding Method

Randomization was performed using a computer-generated random permuted block design, with a block size of 4 and 1:1 ratio. Allocation was sequentially numbered and sealed in opaque envelopes by the primary investigator. A study group member (HY) opened the envelope before induction of anesthesia and prepared the study drug according to the group allocation. The patients, outcome investigators, and surgeons were blinded to group assignment.

### 2.4. Intervention, Anesthesia Protocol, and Perioperative Pain Management

After standard monitoring (non-invasive blood pressure, electrocardiogram, pulse oximetry) and bispectral index monitoring (BIS; Medtronic, Minneapolis, NM, USA), 1.5–2 mg/kg of propofol, 0.8 mg/kg of rocuronium, and 0.05–0.20 μg/kg/min of remifentanil were administered for anesthesia induction. After intubation, anesthesia was maintained using sevoflurane within a BIS level of 40–60. Remifentanil was infused to maintain blood pressure and heart rate within 20% of baseline.

In the nefopam group, 20 mg of nefopam mixed with 100 mL of normal saline was administered intravenously during 15 min immediately after the induction of anesthesia and 15 min before the end of surgery. In the control group, 100 mL of normal saline was administered in the same manner. For postoperative pain control, 0.01 mg/kg of hydromorphone and 1 g of acetaminophen were administered intravenously 20 min before the end of surgery, and intravenous patient-controlled analgesia (IV-PCA; fentanyl 1000 μg diluted with 0.9% saline to make 100 mL of total volume, bolus dose of 1 mL, lockout time of 15 min, and basal infusion rate of 1 mL/h) was also applied for both groups. The tracheal tube was removed after the patient had fully recovered from neuromuscular block and was able to properly obey a command. After extubation, the patient was transferred to the post-anesthesia care unit (PACU) and monitored for approximately 1 h. Pain intensity was measured using a numeric rating scale (NRS; 0 = no pain, 10 = worst pain imaginable), and rescue analgesics (IV hydromorphone 0.01 mg/kg) were allowed with an NRS score ≥ 5. If a patient complained of pain with an NRS ≥ 5 more than 15 min after receiving rescue medication, more hydromorphone (0.3 mg) was administered intravenously.

Postoperative care was performed in the intensive care unit (ICU) during the first night after surgery, and most patients were then transferred to the general ward on the next day. The patients were routinely given 8 mg of hydromorphone orally beginning on the first postoperative day (POD 1). Intravenous hydromorphone (1 mg) or morphine (5 mg) was administered when the NRS score was ≥5. When patients required rescue analgesics more than 3 times per day, ibuprofen, acetaminophen, or tramadol was added orally as routine analgesics. Postoperative nausea and vomiting were treated with 0.3 mg of intravenous ramosetron hydrochloride (Naseron Inj., Boryung Co., Ltd., Seoul, Korea).

### 2.5. Outcome Measurements

The primary outcome was total morphine equivalent consumption during the first 6 h postoperatively. We also evaluated total opioid consumption (the IV-PCA and all rescue opioids) during the PACU stay and for the first 12, 24, and 72 h postoperatively. The dose of IV-PCA opioid was recorded by the intravenous pump device (Accumate 1200, Woo Young Medical, Jincheon-gun, Chungcheongbuk-do, South Korea). All opioid consumption was converted to the intravenous morphine milligram equivalent dose for comparison. The time to first request for rescue analgesics in the ICU or general ward after surgery was also recorded.

Acute postoperative pain was assessed with the NRS score during the PACU stay (the highest value of pain scores reported) and at 6, 12, 24, and 72 h postoperatively. Chronic postoperative pain was evaluated with the Brief Pain Intensity-short form (BPI-SF) questionnaire and the Neuropathic Pain Questionnaire-short form (NPQ-SF) via a phone call visit 3 months after surgery [20,21].

Adverse events, such as nausea and vomiting, dizziness, respiratory depression, and sedation (Richmond agitation sedation scale score of ≤−2 during the daytime), were also evaluated during the first 72 h postoperatively. Aspartate aminotransferase (AST) and alanine aminotransferase (ALT) were monitored until POD 3.

### 2.6. Statistical Analysis

Based on a previous study, we hypothesized that cumulative opioid consumption during the first 6 h postoperatively would be decreased by 20% in the nefopam group [15]. With a two-tailed significance level of 0.05 and a power of 80%, the number of study subjects needed in each group to find statistical differences between the groups was 42. Considering a dropout rate of 15%, 50 patients were included in each group. All patients who were randomized and treated were included in the analysis based on the intention-to-treat principle.

Categorical variables are presented as frequency (percentage), and continuous variables are presented as mean ± standard deviation or median [interquartile range] according to their normality, which we evaluated with the Shapiro–Wilk test. The *t*-test or the Wilcoxon rank sum test was used to compare continuous variables between two groups, and the chi-square test or Fisher’s exact test was used to compare categorical variables. For median differences, 95% confidence intervals (CI) were computed by the 2.5th and 97.5th percentiles of the bootstrap distribution with 1000 bootstrap replications. Bonferroni correction was used for multiple testing. A two-sided, *p*-value < 0.05 was considered statistically significant, and all statistical analysis were performed using Statistical Analysis System (SAS) version 9.4 (SAS Institute, Cary, NC, USA).

## 3. Results

A flow diagram of the study is shown in Figure 1. Between March 2021 and September 2021, 118 patients scheduled for elective VATS for lung cancer were assessed for eligibility. Among those 118 patients, 14 patients who refused to participate in the study and four patients diagnosed with benign disease were excluded. Thus, 100 patients were randomly allocated to the control (*n* = 50) or nefopam (*n* = 50) group. One patient allocated to the nefopam group dropped out because of withdrawal of consent. The operations of 4 patients were converted to thoracotomy, and 11 patients underwent a wedge resection or segmentectomy due to intraoperative changes in the surgical plan. Those 15 patients are included in the analysis based on the intention-to-treat principle. Therefore, 99 patients completed the 72 h follow-up. Of them, 75 patients (37 from the control group, 38 from the nefopam group) completed the 3-month follow-up measures.

The baseline demographics and intraoperative data were comparable between the groups, except that the infusion dose of intraoperative remifentanil was higher in the nefopam group (0.450 [0.300–0.550] mg vs. 0.300 [0.250–0.450] mg; *p* = 0.013; Table 1).

Total opioid consumption during the first 6 h postoperatively did not differ between the nefopam and control groups (19.8 [13.5–25.3] mg vs. 20.3 [13.9–27.0] mg; median difference: −1.55 mg, 95% CI: −6.64 to 3.69; *p* = 0.356). Opioid consumption during the PACU stay and at 12, 24, and 72 h postoperatively was also comparable between the two groups. These data are shown in detail in Table 2. The time between the end of surgery and the first request for rescue analgesia in the ICU or general ward was likewise comparable between the nefopam and control groups (212.5 [104.0–371.0] min vs. 259.5 [142.0–432.0] min; median difference: −47.0 min, 95% CI: −169.00 to 73.95; *p* = 0.302; Table 2). The NRS pain scores at rest during the first 72 h postoperatively did not differ significantly between the groups at any time (Table 3). Likewise, the incidence of adverse events did not differ significantly between the groups. Changes in AST and ALT between the preoperative value and POD 1 or POD 3 did not differ significantly between the groups. The duration of ICU stay and hospitalization did not differ between the groups either (Table 3).

Thirty-seven patients from the control group and thirty-eight patients from the nefopam group responded to a phone call visit 3 months postoperatively. The incidence of chronic pain at 3 postoperative months was 55 and 65% in the nefopam and control groups, respectively (*p* = 0.540). Among the 45 patients who answered that they had persistent pain, the pain scores differed insignificantly between the groups (Table 4). The interference of pain with daily function also differed insignificantly between the groups (online Supporting Information, Appendix A).

In their answers to the NPQ-SF questionnaire, 43 patients (57%) indicated that they had at least one neuropathic pain component, and the incidence of neuropathic pain did not differ significantly between the groups (18/38 [47%] vs. 25/37 [68%], *p* = 0.103). The severity of pain and numbness was also comparable between the groups (online Supporting Information, Appendix A).

## 4. Discussion

In this study, intraoperative nefopam administration did not decrease total opioid consumption or postoperative pain intensity during the first 72 h after VATS for lung cancer. It also did not reduce the incidence of chronic post-surgical pain 3 months after surgery.

Several studies have shown promising results for the multimodal opioid-sparing analgesia of nefopam on acute postoperative pain [15,16,17,18,22,23]. In those studies, the nefopam groups required fewer opioids via IV-PCA or rescue analgesics or showed reductions in pain scores compared with the control groups during the postoperative period. As potential causes of its analgesic effects, those authors suggested triple neurotransmitter inhibition, NMDA receptor antagonism, and the modulation of presynaptic glutaminergic transmission [11,12,13,14].

However, conflicting results have also been reported, especially in surgeries anticipated to cause moderate to severe pain [24,25,26,27]. Cuvillon et al. reported that continuous intravenous infusion of nefopam (120 mg) during the first 48 h after open colectomy did not reduce perioperative opioid consumption and produced no differences in patient satisfaction or adverse events compared with the control group [24]. Eiamcharoenwit et al. administered 30 mg of nefopam before incision, at the end of spine surgery, or at both times and compared the outcomes with placebo. They also found no significant difference in postoperative morphine consumption among the four groups [26]. Other studies have also shown that nefopam had no or limited efficacy on postoperative pain management when it was used as a part of multimodal analgesia [28,29].

Our results are consistent with those studies reporting that the opioid-sparing effect of nefopam is unclear. Many of the studies that demonstrated the analgesic efficacy of nefopam involved surgeries with mild to moderate postoperative pain, such as laparoscopic cholecystectomy, mastectomy, thyroidectomy, and middle ear surgery [22,30,31,32]. 20 mg of intravenous nefopam is comparable to 6–12 mg of intravenous morphine [33], and the median effective dose (ED50) of nefopam for moderate surgical pain was 21.7–28 mg [34,35,36]. We administered 20 mg of nefopam twice during surgery. Although that dose was higher than the ED50 found in previous studies, it might still have been insufficient because the ED50 value was not determined based on thoracic surgery. Although post-surgical pain after VATS is less than that after thoracotomy, it is still severe during the acute postoperative period, and the incidence of CPTPS does not differ from that following thoracotomy [3].

Furthermore, no adequate dose, infusion rate, or duration of nefopam administration has yet been established for postoperative pain control. In previous studies, nefopam was administered in three major ways [11,12]. The first is administration before surgical incision and at the end of surgery, as in our study. The second is continuous infusion for 24–48 h beginning at the end of surgery, and the third is continuous infusion during surgery after an initial administration. The dose of nefopam used in previous studies varied from 20 to 120 mg per day, depending on the type of surgery, operation time, and administration method. However, in those studies, the analgesic effects were different, not uniform.

To prevent the development of CPTPS involving a neuropathic pain component, perioperative pain control is very important [1,3,4,5]. The action of nefopam on the glutaminergic pathway was proven in in vitro studies, and its antiallodynic and antinociceptive effect on neuropathic pain was also demonstrated in in vivo animal studies [37]. Ok et al. reported that additive nefopam in IV-PCA reduced neuropathic pain after percutaneous endoscopic lumbar discectomy [18]. In this study, however, nefopam did not reduce the incidence of acute, chronic or neuropathic pain after VATS for lung cancer. This result could reflect an inadequate dose of nefopam. One study in laparoscopic colectomy also reported that 20 mg of nefopam did not reduce acute or chronic postoperative pain [38]. Those authors suggested that a low dose of nefopam caused negative preemptive analgesic results, which might not be enough to prevent nociceptive transmission and central sensitization for moderate to severe pain [38]. On the other hand, a study in breast surgery reported that 20 mg of nefopam administered before surgical incision reduced the use of rescue analgesics and lowered the incidence of chronic postoperative pain [30].

In this study, five patients who complained of moderate to severe pain (NRS score ≥ 5) 3 months after surgery were all in the control group. Although there was no statistically significant difference between the groups, that result could suggest that nefopam has a potential role in attenuating severe pain in CPTPS. Further studies will be needed to clarify the effect of nefopam on chronic postoperative pain.

A characteristic finding of this study is that the dose of remifentanil infused during surgery was significantly higher in the nefopam group. We attribute that finding to the tachycardia effect of nefopam [11]. However, a study reported that nefopam had analgesic efficacy after laparoscopic gastrectomy and that intraoperative remifentanil consumption was lower in the nefopam group than in the control group [15]. The mean infusion rate of the control group in that study was 0.13 ± 0.06 μg/kg/min. As an explanation for their finding, those authors suggested that nefopam might be an NMDA receptor antagonist and thereby prevent remifentanil-induced hyperalgesia. In our study, on the other hand, the mean infusion rate in the nefopam and control groups was 0.07 ± 0.03 μg/kg/min and 0.05 ± 0.02 μg/kg/min, respectively. Only one patient in the nefopam group was given a dose of remifentanil greater than 0.1 μg/kg/min, which could induce remifentanil-induced hyperalgesia [39]. Therefore, it is unlikely that the difference in remifentanil infusion between the groups affected postoperative opioid consumption.

This study has several limitations. First, the patients in this study were not completely blinded. Although they did not know which group they were in during the acute postoperative period, they could have known their group if they read their medical records later. Second, we only evaluated postoperative pain intensities at rest. Assessing pain scores during coughing or movement would have been more appropriate. Third, we could not evaluate the incidence of tachycardia and sweating during the perioperative period, and they are known to be frequent adverse effects of nefopam. However, it was difficult to evaluate the occurrence of tachycardia during surgery because many factors can induce tachycardia during VATS. Fourth, the chronic pain evaluation was conducted by telephone, rather than in face-to-face interviews. Finally, the sample size of this study was not determined by the incidence of CPTPS. Furthermore, only 37 patients in the control group and 38 patients in the nefopam group were evaluated chronic pain at postoperative 3 months due to the follow-up loss. This sample size was rather small to demonstrate the incidence of chronic pain or compare it between the two groups.

## 5. Conclusions

In conclusion, intraoperative nefopam administration did not decrease total opioid consumption or postoperative pain intensity during the first 72 h after VATS for lung cancer. It also did not reduce the incidence of chronic post-surgical pain 3 months after surgery, compared with the control group. Further studies are required to elucidate the potential role of nefopam in multimodal analgesia for patients undergoing VATS.

## Figures and Tables

**Figure 1 jcm-11-04849-f001:**
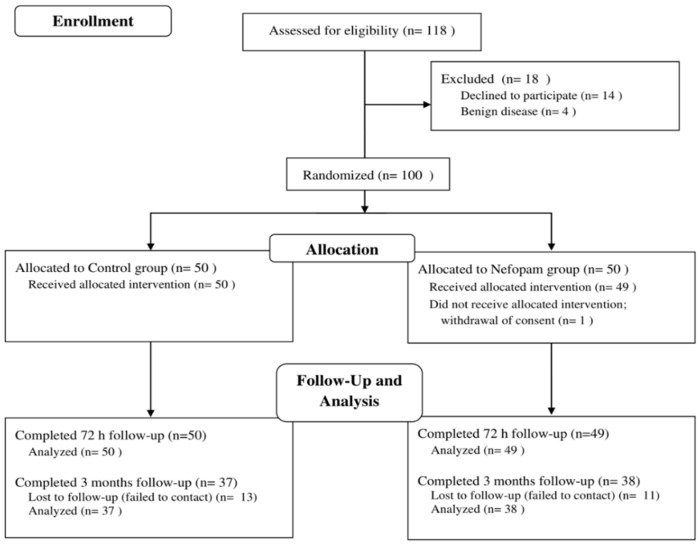
CONSORT flow diagram.

**Table 1 jcm-11-04849-t001:** The baseline characteristics and perioperative clinical data.

	Control Group (*n* = 50)	Nefopam Group (*n* = 49)	*p*-Value
Age, year	59.5 [55.0, 63.0]	59.0 [53.0, 63.0]	0.606
Males/Females, *n*	17 (34)/33 (66)	21 (43)/28 (57)	0.484
Weight, kg	62.7 [55.0, 67.6]	64.5 [56.5, 69.9]	0.566
Height, cm	160.6 ± 7.8	162.8 ± 8.4	0.181
Body mass index, kg/m^2^	24.1 [21.9, 26.4]	24.1 [22.7, 24.9]	0.740
Smoking, *n*	16 (32)	21 (43)	0.573
Hypertension, *n*	10 (20)	10 (20)	>0.999
Diabetes mellitus, *n*	5 (10)	6 (12)	0.972
ASA physical class (I/II/III), *n*	5/42/3	10/36/3	0.383
Type of surgical incision, *n*			0.362
VATS	49 (98)	46 (94)	
Thoracotomy	1 (2)	3 (6)	
Type of surgery, *n*			0.189
Lobectomy	47 (94)	41 (84)	
Wedge resection or Segmentectomy	3 (6)	8 (16)	
Operator (1/2/3/4/5/6/7), *n*	7/9/5/17/0/1/11	11/11/4/12/2/4/5	0.257
Duration of anesthesia, min	157.5 [135.0, 197.0]	166.0 [143.0, 193.0]	0.669
Duration of operation, min	104.5 [84.0, 145.0]	109.0 [88.0, 148.0]	0.583
Amount of intraoperative remifentanil administration, μg	300.0 [250.0, 450.0]	450.0 [300.0, 550.0]	0.013 ^1^

Values are *n* (%), mean ± standard deviation or median [interquartile range].^1^
*p*-value < 0.05; ASA: American Society of Anesthesiologists, VATS: video-assisted thoracoscopic surgery.

**Table 2 jcm-11-04849-t002:** Postoperative analgesic outcomes between the groups.

	Control Group (*n* = 50)	Nefopam Group (*n* = 49)	Median Difference 95% CI	*p*-Value
**Total opioid consumption (mg)**				
**6 h postoperatively ^1^**	20.3 [13.9, 27.0]	19.8 [13.5, 25.3]	−1.55 [−6.64, 3.69]	0.356
During PACU stay	8.9 [6.9, 10.1]	7.4 [6.1, 9.4]	−1.45 [−2.76, 0.07]	0.196 ^2^
12 h postoperatively ^3^	31.6 [26.3, 39.7]	31.5 [22.7, 41.0]	−0.04 [−7.93, 8.10]	>0.999 ^2^
24 h postoperatively	58.5 [48.8, 78.7]	63.0 [48.8, 83.5]	4.45 [−8.96, 16.29]	>0.999 ^2^
72 h postoperatively	120.0 [85.0, 166.4]	130.6 [109.0, 178.0]	10.57 [−9.53, 33.50]	0.618 ^2^
Time to first rescue analgesia (min)	259.5 [142.0, 432.0]	212.5 [104.0, 371.0]	−47.00 [−169.00, 73.95]	0.302

Values are presented as morphine milligram equivalent doses and median [interquartile range]. For median difference, 95% CI are computed by the 2.5th and 97.5th percentiles of the bootstrap distribution by 1000 bootstrap replications.; ^1^ Primary outcome: amount of morphine equivalent consumption includes both IV-PCA and all rescue opioids.; ^2^ Bonferroni’s method was used for multiple comparisons at four time points; during PACU stay, 12, 24 and 72 h postoperatively.; ^3^ For the postoperative 12 h readings, *n* = 98 due to 1 follow up loss of PCA data; *n* = 49 for both groups; CI, confidence interval; PACU, post-anesthesia care unit.

**Table 3 jcm-11-04849-t003:** NRS pain scores at each time point and postoperative clinical outcomes between the groups.

		Control Group (*n* = 50)	Nefopam Group (*n* = 49)	Median/Risk Difference 95% CI	*p*-Value
NRS pain score during 72 h postoperatively					
PACU stay ^1^		6.0 [5.0, 7.0]	5.0 [3.0, 6.0]	−1.0 [−2.0, 0.0]	0.355 ^2^
6 h postoperatively		3.0 [3.0, 4.0]	3.0 [3.0, 4.0]	0.0 [−1.0, 1.0]	>0.999 ^2^
12 h postoperatively		3.0 [3.0, 5.0]	3.0 [3.0, 4.0]	0.0 [−1.0, 0.0]	>0.999 ^2^
24 h postoperatively		3.0 [2.0, 5.0]	3.0 [2.0, 4.0]	0.0 [−1.0, 1.0]	>0.999 ^2^
72 h postoperatively		2.0 [2.0, 3.0]	2.5 [1.0, 3.0]	0.5 [0.0, 1.0]	>0.999 ^2^
Incidence of adverse events					
PONV		33 (66)	34 (69)	3.4 [−15.0, 21.8]	0.884
Dizziness		24 (48)	19 (39)	−9.2 [−28.7, 10.2]	0.470
Desaturation		8 (16)	8 (16)	0.3 [−14.2, 14.8]	>0.999
Sedation		1 (2)	2 (4)	2.1 [−4.7, 8.9]	0.617
Changes in AST and ALT at POD1 and POD3 ^2,3^					
AST	POD1	3.0 [0.0, 6.0]	5.0 [0.0,11.0]	2.0 [−1.0, 7.0]	0.144
	POD3	0.0 [−3.0, 5.0]	0.5 [−4.5, 4.0]	0.5 [−3.0, 3.0]	>0.999
ALT	POD1	−1.0 [−7.0, 1.0]	−1.0 [−5.0, 4.0]	0.0 [−3.0, 4.0]	0.864
	POD3	−2.0 [−7.0, 0.0]	−2.0 [−8.0, 3.5]	0.0 [−4.0, 3.0]	>0.999
Duration of ICU stay (min)		1140.0 [1020.0, 1280.0]	1177.0 [1050.0, 1339.0]	37.0 [−35.5, 50.0]	0.283
Duration of hospitalization (day)		6.0 [5.0, 7.0]	6.0 [5.0, 7.0]	0.0 [−1.0, 1.0]	0.904

Values are *n* (%) or median [interquartile range]. For median difference, 95% CI are computed by the 2.5th and 97.5th percentiles of the bootstrap distribution by 1000 bootstrap replications.; ^1^ Highest NRS score during PACU stay; ^2^ Bonferroni’s method was used for multiple comparisons.; ^3^ Changes in AST and ALT were calculated by subtracting the preoperative value from POD1 or POD3 value. CI, confidence interval; NRS, numeric rating scale; PACU, post-anesthesia care unit; PONV, postoperative nausea and vomiting; AST, aspartate aminotransferase; ALT, alanine aminotransferase.; ICU, intensive care unit.

**Table 4 jcm-11-04849-t004:** The short form of brief pain inventory on 3 months after surgery.

	Control Group (*n* = 37)	Nefopam Group (*n* = 38)	Median Difference 95% CI	*p*-Value
Presence of pain	24 (65)	21 (55)		0.540
Pain intensity ^1^				
Worst pain	3.00 [2.00, 4.00]	2.00 [1.00, 3.00]	−1.0 [−2.0, 1.0]	0.150
Least pain	0.00 [0.00, 0.00]	0.00 [0.00, 0.00]	0.0 [0.0, 0.0]	0.189
Average Pain	1.00 [0.75, 2.00]	1.00 [0.00, 2.00]	0.0 [−2.0, 1.0]	0.299
Current Pain	0.00 [0.00, 1.25]	0.00 [0.00, 0.00]	0.0 [−1.0, 0.0]	0.272
Pain interference with daily activities ^2^	0.00 [0.00, 0.00]	0.00 [0.00, 0.00]	0.0 [0.0, 0.0]	NS

Values are *n* (%) or median [interquartile range]. For median difference, 95% CI are computed by the 2.5th and 97.5th percentiles of the bootstrap distribution by 1000 bootstrap replications.; Each item (except for presence of pain) is rated on a numeric rating scale from 0 (no pain) to 10 (worst) or from 0 (no interference) to 10 (interferes completely). ^1^ Pain intensity scores and interference items were analyzed only among the patients that they have current pain (control group, *n* = 24; nefopam group, *n* = 21). ^2^ The median (IQR) value of all interference items was the same as above, and details are attached as supplementary data.; CI, confidence interval; NS: not significant.

## Data Availability

The data presented in this study are available on request from the corresponding author.

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
