# Peer review of "The Lack of Analgesic Efficacy of Nefopam after Video-Assisted Thoracoscopic Surgery for Lung Cancer: A Randomized, Single-Blinded, Controlled Trial"

_jcm, 2022, doi:10.3390/jcm11164849_

Round 1

Reviewer 1 Report

The authors evaluated the analgesic efficacy of intraoperative nefopam on postoperative pain after video-assisted thoracoscopic surgery (VATS) for lung cancer in a cohort of 99 patients. The study was designed as a prospective, single-blinded, randomized controlled trial.
The authors concluded that intraoperative nefopam did not decrease acute postoperative opioid consumption or pain intensity, nor did it reduce the incidence of chronic pain after VATS.
Overall, well written and designed study. However, I have several, mostly minor, objections to improve the quality of the study:

1.    Before the aim of the study in abstract the authors should add a sentence or two in regards to the investigated topic. Also, I would suggest to the authors to avoid a form: 'we evaluated'… I would suggest: The analgesic efficacy of intraoperative nefopam on postoperative pain after video-assisted thoracoscopic surgery (VATS) for lung cancer was used. Please revise through the text and in conclusion!

2.    As this was a prospective, single-blinded, randomized controlled trial it should be registered at ClinicalTrials.gov, or another similar platform. Please add a reference.

3.    Paragraph 2.2. – 'VAT' should be replaced with 'VATS'

4.    Paragraph 2.6. – Please add a full explanation of the abbreviation 'SAS'.

5.    At the end of paragraph 2.6. there are two full stops, please delete one.

6.    Calculated sample size was 42 patients per group. At the end of the study in intervention and control groups 37 and 38 patients completed study, respectively. This should be addressed in limitations of the study.

Author Response

We appreciate for kind and detailed review to the manuscript. We have carefully examined and revised the manuscript according to your recommendations. Please see the attachment.

Reviewer 2 Report

Thank you for submitting the manuscript. I read your paper with great interest and attention. The issue of chronic postoperative pain is very important, little investigated and with great repercussions on the quality of life of patients. I suggest some small revisions that I believe could improve the quality of your manuscript.First of all, seeing that the results of using nefopam are disappointing, I ask you to insert the results of your search in the title so as to immediately direct the reader on the meaning of your search.In my opinion, the introduction is too superficial: it is necessary to better develop some themes that are only hinted at: the rationale for using nefopam, why you have chosen it, the incidence of acute and chronic pain after thoracic surgery and what can be the determinants . for this I suggest some references that I would like you to read and use as a bibliography:

doi: 10.1093/pm/pnw230.

  • DOI: 10.3390/children8080642
  •  

doi: 10.31616/asj.2021.0337

doi: 10.1053/eujp.2001.0225.

I hope these comments are helpful to you.

Kind Regards

Author Response

(The authors gave the same response as above.)
